# Seeding food security: Overcoming barriers to quality potato seed adoption among smallholders in Kenya

Margaret N. Mwangi[1], Wilckyster N. Nyarindo[1], Marther W. Ngigi[2], Hezron N. Isaboke [1]*

**1** Department of Agricultural Economics & Extension, University of Embu, Embu, Kenya, **2** Department of Agricultural Sciences, Machakos University, Machakos, Kenya

\* isaboke.hezron@embuni.ac.ke

## Abstract

Access to quality seed potatoes is essential for boosting productivity and food security among smallholder farmers. However, the adoption and use of these seeds remain low in Kenya, thus constraining yield potential and dietary diversity. This study investigates the adoption, level of use, and impact of quality seeds on household food security among smallholder farmers in Kenya. Using a semi-structured questionnaire, data were collected from 541 households through a multi-stage sampling approach. The Heckman two-stage model estimated factors influencing adoption and usage intensity, while the propensity scores matching model assessed the impact of quality seed use on household food security. The findings indicate that about 44% of farmers used quality seeds, with significant barriers including unavailability, limited accessibility, and high cost of these seeds. Key drivers of adoption and usage intensity included the level of education, total land size, household income, access to extension, digital information, credit access, membership in a farmers' organization, access to subsidies, livestock portfolio, and geographical location. Adopting quality seeds enhanced household food security, as evidenced by significantly higher household dietary diversity scores (HDDS) and food consumption scores (FCS). This improvement was largely attributed to increased intake of nutrient-dense foods, particularly animal-source proteins, fruits, and vegetables. These results underscore the need for policies that enhance local certified seed multiplication and distribution to increase access and reduce the cost of quality seeds. Supporting the adoption of quality seeds can substantially improve agricultural productivity and strengthen food and nutritional security.

## Introduction

Food security remains one of the most urgent global development challenges, with Africa identified as the least food-secure region worldwide [1]. Despite global efforts

**Data availability statement:** The dataset underlying this study is publicly available and can be accessed via the following DOI:https://doi.org/10.6084/m9.figshare.31331608.

**Funding:** This work was supported by a PhD scholarship awarded to M.N.M under the Food Systems Resilience Project (FSRP), implemented by the Ministry of Agriculture and Livestock Development, Government of Kenya (IDA Credits 7327-KE and 7328-KE; Project ID No. P177816). Additional funding was provided by the Machakos University Internal Research Grant 2023/2024 under the project entitled "Creating Synergy in the Irish Potato Value Chain for Quality Seed, High-Quality Potatoes and Market Development in Kenya," awarded to M.W.N. The funders had no role in study design, data collection and analysis, decision to publish, or preparation of the manuscript.

**Competing interests:** The authors have declared that no competing interests exist.

and progress in recent years to combat malnutrition, many countries are still far from eradicating hunger and food insecurity by 2030 [2]. A joint annual report by [3] revealed that, between 648 and 720 million people- approximately 8% of the world's population faced hunger, and 2.37 billion people experienced moderate or severe food insecurity in 2024. Sub-Saharan Africa had the highest number of affected individuals compared to other regions [3,4]. Even more concerning is that approximately 46 million people in East Africa are severely food insecure, and 12 million people face acute malnutrition [5]. Estimates indicate that the prevalence of food insecurity and malnutrition in Kenya is high, with approximately 2.15 million Kenyans facing acute food insecurity and 18% of children aged 5–59 months suffering from chronic malnutrition [6].

Food insecurity and malnutrition have detrimental social and economic effects, decreasing individuals' health and productivity, contributing to around 8% losses in Kenya's GDP, and accounting for roughly 45% of all child mortality cases reported annually [7]. Addressing food security is, therefore, essential not only in achieving global goals such as Sustainable Development Goals (SDGs): SDG targets 2.1 and 2.2 (ending hunger and malnutrition, respectively), but also in fulfilling Kenya's Fourth Medium Term Plan (MTP IV, 2023–2027) and Vision 2030. This emphasizes the vital need for agricultural interventions that increase yields, improve food security, and strengthen resilience. Strengthening seed systems is particularly critical, as they form the foundation for sustainable productivity growth, access to improved varieties, and long-term food system resilience.

Agricultural innovations, especially using quality seeds, present a promising avenue to increase agricultural productivity and food security [8,9]. High-quality seeds outperform conventional seeds in terms of genetic and physiological traits as well as performance [10]. Existing literature suggests that using high-quality seed from a well-functioning seed system, combined with good agronomic practices, can close the yield gap, increase pest and disease resistance, and reduce costs for agrochemicals, ultimately boosting farmers' income and food security [11]. A recent study by [12] found that Ecuador farmers who used quality seed potatoes achieved a higher yield of 25 tons/ha, which was 52% greater than that obtained without higher quality seeds. Similarly, [13] observed that increasing the availability and use of quality seed among Ethiopian farmers improved their yields by approximately 30–50%. Okello et al. [14] reported a yield increase of approximately 2.97–9.52 tons/ha among farmers who used quality seeds in Kenya.

While there is extensive literature on the role of quality seeds in boosting agricultural productivity [12,14–17], only a handful have focused on their impact on improving dietary diversity- a vital aspect of food [9,18]. Beyond improving quantity, quality seeds can also boost dietary diversity through multiple pathways. First, high-quality seeds allow farmers to achieve higher yields on small plots, freeing up land and other resources for growing different crops. This broadens access to healthier and more diverse diets [18]. For example, a study by [19] found that Malian farmers using improved sorghum seeds had greater access to diverse diets than those using local seeds. Secondly, quality seeds increase crop yields, enabling

farmers to produce surplus for the market, raising household income, and allowing them to buy a variety of food [2,20]. Evidence suggests that consuming a range of food groups supports balanced intake of all essential nutrients needed for good health [21]. Thirdly, quality seeds mature faster, allowing farmers to rotate or intercrop them with other crops. This improves access to diverse foods across seasons and reduces seasonal food shortages caused by biotic and abiotic factors [22,23]. Fourthly, quality seeds are resistant to diseases such as bacterial wilt and late blight, and are drought-tolerant [24]. This minimizes episodes of crop failure and stabilizes yields by imparting some resilience to biotic and abiotic stressors [22].

Despite the aforementioned benefits of quality seeds, their adoption among smallholder farmers remains strikingly low, with significant variation in usage levels in many countries in sub-Saharan Africa, including Kenya [10]. Estimates indicate that the use of quality seeds among smallholder farmers in SSA is below 40% [25] and remains very low for root and tuber crops in East Africa, where the formal seed system remains undeveloped [26,27]. Smallholder farmers, therefore, resort to using local or indigenous seed varieties, which are vulnerable to abiotic and biotic stresses, leading to low yields [28].

Therefore, understanding the factors and challenges that limit the wider adoption of quality seeds is essential in designing and implementing programs and policies to improve agricultural productivity. Empirical studies highlight that farmers will only adopt quality seeds if they are available, accessible, affordable, and if farmers are aware of them [29,30]. However, smallholder farmers face multiple structural and physical barriers in accessing quality seeds. For example, a study by [31] found that complex procurement issues and low supply of clean seed potatoes prevented Kenyan farmers from adopting them. Okello et al. [32] discovered that the higher seed cost, mainly due to increased transaction costs, discouraged adoption among smallholder potato farmers. Adoption decisions are also influenced by farmers' perceptions of seed quality [33], socio-economic status [34], and institutional factors [35,36]. Additionally, a lack of information about seed sources and limited awareness of the benefits of quality seeds contribute to their low uptake [37]. While these factors help explain adoption decisions, it is still unclear whether they also affect the intensity and sustained use of quality seeds. Furthermore, it remains uncertain whether quality seeds, especially seed potatoes, can substantially help achieve food and nutritional security at the household level.

This study contributes to the existing literature by evaluating the adoption patterns, use intensity, and food security impacts of quality seed potatoes among smallholder potato farmers in Kenya. Specifically, it addresses the following interlinked research questions: What challenges and constraints hinder farmers' access to quality seed potatoes? What factors influence the adoption and intensity of adoption of quality seeds? How does using quality seed potatoes affect household food and nutritional security? The findings from this study will provide critical insights for designing targeted policies that address the barriers farmers face in adopting and effectively using quality seed potatoes, thereby contributing to increased productivity and improved household food security.

We focus on potatoes since the crop presents an opportunity through which food security can be achieved [38]. The crop provides food for millions of smallholder farmers in Kenya [28]. Its ability to mature faster, as well as give more returns to land and labor, makes it a potential crop for addressing food insecurity [39]. In addition, the crop has high climatic adaptability owing to its ability to grow in high-altitude areas where most cereals do not perform well [40]. Despite the vast potential for the crop, its output per unit area is low, averaging 8 tons/ha against a potential of over 30 tons/ha with the best production technologies [40]. The low productivity is attributable to many factors, with inadequate use of quality seeds taking center stage [37]. Most smallholder farmers (about 76%) rely on informal seed systems to procure their planting materials [28,41]. Under this system, farmers either retain their home-saved seeds, obtain them from their neighbors, or purchase them from local markets [42]. The quality of seeds obtained from these informal sources is poor and susceptible to soil-borne diseases [43]. Moreover, farmers often reuse these seeds for multiple cropping cycles without replenishing seed stocks, which causes seed degeneration due to pathogen buildup, leading to a significant drop in potato yields [28,35].

## Conceptual framework

The study is based on the premise that seed systems can boost food security and nutrition by enhancing access to high-quality seeds. The seed systems can either be formal, informal, or semi-formal, with substantial diversity in seed sources, method of seed multiplication, and distribution pathways [35]. In informal seed systems, farmers obtain seeds through various channels such as own-saved, neighbors, local markets, and cooperatives, often without government control [44]. More often, farmers select small seeds that cannot be sold as seeds [45]. Using seed potatoes from these informal sectors for multiple generations without replenishing them can cause a substantial reduction in the quantity and quality of potatoes due to the accumulation of diseases and pests [28]. The semi-formal entails local community-based seed production initiatives, such as quality-declared seeds, with less stringent quality control measures [44]. Conversely, in the formal seed system, the production, distribution, and marketing of seeds of known quality and varieties are guided by established standards and regulations [46]. This system is mainly responsible for producing and distributing improved or modern varieties in the form of quality seeds, which have higher resistance to diseases, drought tolerance, and mature faster, leading to higher yields [44].

We conceptualize four pathways through which the adoption and use of these quality seeds can improve food and nutritional security, as depicted in Fig 1. The first is the **production pathway**, implying that using these seeds increases yield among smallholder farmers. This translates to higher household food availability, resulting in more diversity in meals

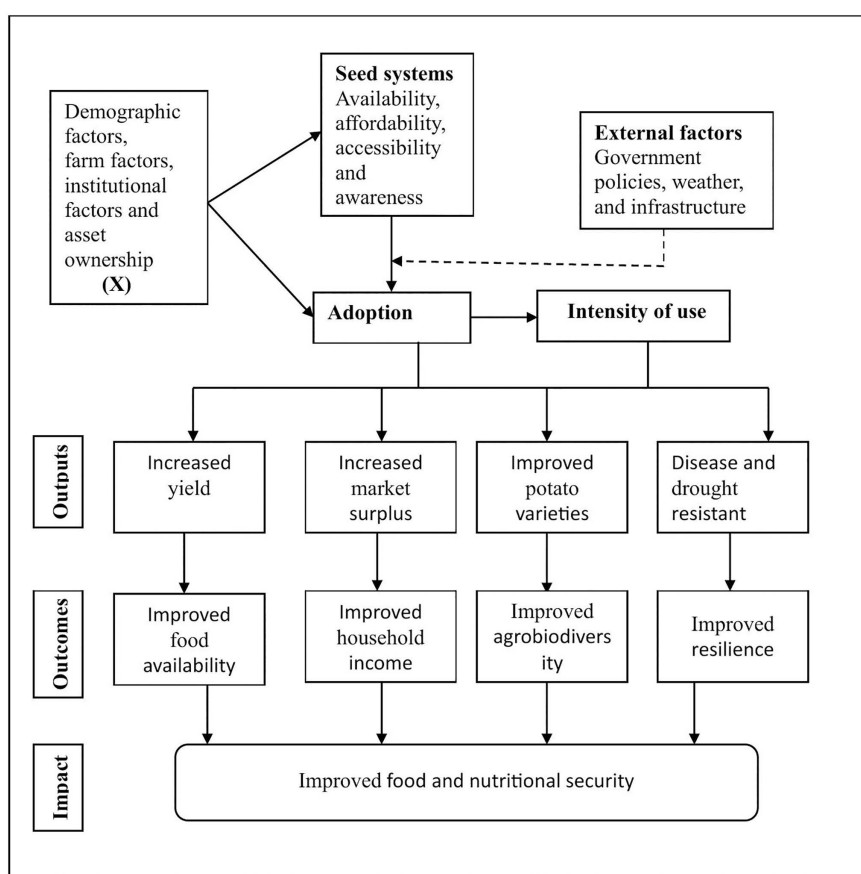

**Fig 1. Conceptual framework illustrating the pathways through which quality seed adoption improves food and nutritional security.**

and, thus, higher dietary diversity [14]. The second is the **income pathway**, suggesting that using quality seeds enables farmers to increase their market surplus, thus improving their household income through the sale of potato surplus, and enabling households to buy other food items, leading to increased dietary diversity [47]. The third is through the **agrobiodiversity pathway**. Accessing quality seeds enables farmers to grow diversified potato varieties with varied preparation options, some for making chips and crisps, others for roasting, and others for boiling and mashing, thus enhancing dietary diversity at the household level. The fourth is the **resilient pathways**, which implies that using quality and improved seed varieties enables households to adapt to climate change. Improved varieties like those developed by the International Potato Center (CIP) are drought-tolerant, resist diseases, and mature faster (Unika, Sherekea, Wanjiku, and Nyota), allowing for more flexibility in crop planting and harvesting [24,45]. This leads to consistent food availability and access, sustaining household dietary diversity throughout the season.

Nevertheless, adoption and use of quality seeds, along with the expected outcomes, may not be homogeneous among smallholder farmers. Factors like demographics, institutional support, farm features, and location can affect both the decision to adopt and the level of use of quality seeds [36]. For example, younger and well-educated farmers with larger farms are more likely to use quality seeds [48]. Likewise, farmers with better access to credit, agricultural training, digital information, extension services, and social networks are more likely to adopt and intensify the use of quality seeds, as these institutional services improve their financial capacity, technical knowledge, decision-making ability, and awareness of the benefits of quality seeds [49]. Other factors outside farmers' control (external factors) can also influence the decision to adopt quality seeds. For instance, government policies such as subsidies, certifications, infrastructure (roads, storage facilities, and markets), and agro-ecological conditions may not be within farmers' control but can significantly impact the adoption of these seeds [50]. Recognizing this complexity, the study examines the link between adopting and using quality seeds and their impact on household food and nutritional security (dietary diversity and food consumption frequency).

## Materials and methods

### Study area

The study was conducted in Nyandarua, Nakuru, and Meru Counties between 4th June and 3rd August 2024. These counties are located in the Central, Rift Valley, and Eastern Regions of Kenya, respectively, representing a diverse agronomic environment. Several programs and initiatives are also being implemented in these counties to enhance potato production. Among the implemented programs are the establishment of cold storage facilities, subsidized fertilizers, and cooperative formations to link farmers to output markets. However, potato production in the three counties remains below its potential, primarily due to low usage of quality seeds, potato pests and diseases, and post-harvest losses [32].

### Sampling design and technique

A multi-stage stratified sampling method was employed to systematically select counties, sub-counties, and farm households for the study. In the initial stage, three counties were chosen based on their dominance in Irish potato production, the concentration of potato seed production units, and collective and contractual arrangements. The second stage involved purposive sampling of three sub-counties from each county where potato was mainly grown to ensure diverse representation. The sub-counties included Buri, Central Imenti, and South Imenti in Meru County; Molo, Njoro, and Kuresoi North in Nakuru County; and Olkarau, Kinangop, and Oljororok in Nyandarua County. In the third stage, three wards within each selected sub-county were randomly chosen to ensure geographical representation. The final stage entailed random sampling of farmers from selected wards, following the list of farming households obtained from ward agricultural officers.

**Ethical consideration and consent**

Ethical approval for this study was obtained from the Chuka University Institutional Research Ethics Committee (Chuka University, Kenya) under approval number NACOSTI/NBC/AC-0812, dated 25 May 2024, and registered with the National Commission for Science, Technology, and Innovation (NACOSTI) (license number NACOSTI/P/24/32824). The Committee approved the use of informed verbal consent because written consent was not feasible due to low literacy levels among many respondents in the study area. Prior to each interview, the purpose of the study was explained to the participants in their local language, and verbal consent was obtained and recorded in the semi-structured question-naire by the enumerator. Participation was voluntary, and respondents were assured of their right to withdraw at any time. Privacy, anonymity, and confidentiality were strictly maintained throughout the survey. All respondents were 18 years of age or older.

**Data collection**

The survey questionnaire was programmed using Kobo Toolbox to improve efficient data collection, timely monitoring, and validation in the field. Research assistants received comprehensive data collection training to ensure high-quality and accurate results while adhering to ethical research standards. The questionnaire was piloted in Central Imenti Sub-County, which shared socio-economic and agroecological conditions similar to those of the targeted study area. The database captured household and farm characteristics, seed system information, challenges and barriers to seed access, food security status, and farmers' adoption behavior. One focus group discussion (FGD) in each county comprising eight participants (three adult males, three adult females, and two youth farmers) and six key informant interviews (KIIs) with diverse stakeholders along the potato value chain, including Sub-County agricultural officers (SCAO), Ward extension agents, lead farmers, agro-dealers, policy makers and local seed multipliers were conducted to complement the quantita-tive data.

**Measurement of key variables**

**Adoption and intensity of use of quality seeds.** Quality seed refers to planting material that meets established standards for germination, physical purity, and genetic identity and is free from diseases and pests, capable of producing high yields [25,28,41]. In this study, quality seeds refer to potato seeds obtained from recognized formal or semi-formal sources, including certified seed, seeds produced through rapid multiplication technologies (RMT), and known-source seeds acquired from recognized seed multipliers. Farmers who reported using any of these seed types were classified as adopters of quality seed, while those who used their own saved seed, borrowed from friends, or obtained them from local markets were classified as non-adopters of quality seed. For descriptive statistics and econometric analysis, adoption was therefore captured as a binary variable: a value of 1 was assigned to farmers using quality seeds (adopters) and 0 to those who did not (non-adopters). The intensity of agricultural technology adoption indicates the extent of technology use among adopters [51]. This study measured the intensity of quality seed adoption as the total land area in acres allocated to quality seeds among adopting households.

Partial adoption suggests the technology is underutilized, while increased use of the existing technology is preferred over introducing new ones [52]. Therefore, it is crucial to understand the factors influencing both the adoption and inten-sity of adoption of the technology. We expect farmers using quality seeds to obtain higher yields, improved potato quality, greater resistance to pests, diseases, and drought, and ultimately improve their food security compared to non-adopters.

**Household food security.** The Food and Agriculture Organization (FAO) of the United Nations defines food security as a situation where all people, at all times, have reliable physical, social, and economic access to adequate, safe, and nutritious food that meets their dietary needs and preferences for an active and healthy life. Multiple validated household-level indicators have been proposed and applied to measure food security, including the household food insecurity

experience scale (HFIES), the household hunger scale (HHS), the coping strategies index (CSI), the months of adequate household food provisioning (MAHFP), the household dietary diversity score (HHDS), and the food consumption score (FCS) [5,53,54]. The choice of indicator depends on the measured food security dimension, available resources, time frame, and study context [54]. However, it is worth noting that these indicators have their strengths and limitations, and the combination of indicators is recommended over using a single indicator [53].

This study used the HDDS and the FCS to measure food security. Both indicators are qualitative measures of household dietary patterns that reflect household access to a variety of foods and serve as a reliable proxy for food security [55]. The two indicators are easy to implement and require fewer resources than other methods [5]. They also provide a wider view of dietary quality and medium-term food access [56]. FCS is a frequency-weighted dietary diversity score that reflects household food dietary variation, consumption frequency, and nutritional significance over a 7-day recall period [57]. The FCS has been widely utilized in the World Food Programme (WFP) and in studies evaluating the welfare outcomes of agricultural technology [58,59]. The FCS was calculated by asking respondents to report the number of days in the past seven days they consumed each of the eight standard food groups (main staples, pulses, vegetables, fruits, meat, dairy products, fats/oils, and sugar). The number of days for each food group was then multiplied by the corresponding weights based on nutritional significance. The FCS was subsequently determined as the total sum of the weighted scores across all food groups.

Conversely, HDDS measures the economic ability of a household to access a variety of foods that meet all members' nutritional requirements [56]. HDDS was calculated based on the number of FAO's 12 food groups (Cereals & grains; Roots & Tubers; pulses, legumes, nuts; vegetables; fruits; meat, poultry, offal/ organ meat; fish/seafood; eggs; milk & dairy products; oil/fat; sugar/sweets; and condiments/spice) consumed in the past 24 hours, and offered insight into dietary variety. The respondents were asked whether they consumed one item from each food group, and the responses were recorded as 1 for yes and 0 for no. The total dietary diversity score, ranging from 0 to 12, was then calculated by summing the scores for all food groups. A higher score indicated greater dietary diversity, a proxy for improved nutrient adequacy [60].

The FAO's 12 food groups were then categorized into four nutrient-dense categories for descriptive statistics. These categories included animal-protein sources (meat, fish, eggs, and milk), which were coded as 1 if any of these foods were consumed and 0 otherwise, then summed to create a score ranging from 0 to 4. The same approach was applied to energy-food sources (cereals, roots & tubers, oil/fats, and sugars); micronutrient-rich foods (vegetables and fruits) ranged from 0 to 2, and plant protein sources (pulses, nuts, or legumes) ranged from 0 to 1. The mean scores for each nutrient-dense food category were compared across counties and between adopters and non-adopters of quality seeds using one-way ANOVA with Tukey post-hoc tests to examine the significant differences in household consumption patterns across the counties.

## Analytical framework

Quantitative and qualitative analytical methods were used to comprehensively understand the factors influencing adoption, usage intensity, and the impact of quality seed potatoes on household food security. The study also explored farmers' challenges and constraints in accessing quality seeds, along with available opportunities. The quantitative data were exported to the STATA package 15 for cleaning and analysis. A two-sample t-test was used to compare the mean values of continuous variables (e.g., age, farm size, household size) between adopters and non-adopters of quality seeds. Pearson's chi-square test assessed associations between categorical variables and adoption status. Additional inferential statistics, such as Heckman two-stage and propensity score matching (as described in the next section), were employed to identify the determinants and intensity of adoption and the causal effect of quality seeds on household food security. The findings were validated by the qualitative data obtained from FGD and KIIs.

## Heckman two-stage model

The study examined both the adoption and intensity of adoption of quality seeds following a two-step approach. In the first stage, farmers decide whether to adopt quality seeds. In the second stage, they determine the intensity of use after adoption. Selection bias may arise because potato farmers might self-select to adopt based on specific characteristics. The Heckman two-stage, Tobit, and double hurdle models have been commonly used in the literature to determine the adoption and intensity of adoption of technology. Nevertheless, each model has its strengths and weaknesses, and the choice of any of the models over the other depends on the study's objective and the nature of the data being analyzed [61]. The Tobit model is used when the dependent variable is censored. The model is criticized because it assumes that the drivers of the decision to adopt are similar to those that affect the intensity of adoption [62]. However, this may not be the case in our study because the decision to adopt and intensity of adoption of these seeds is made sequentially, implying that the factors that may influence adoption may vary from those that influence the intensity. It also treats zero values in the second equation (intensity) as a corner solution [61]. However, in our case, the observed zeros are due to the discrete choice of not adopting quality seeds.

On the other hand, the double hurdle and Heckman model are the best options for this study. However, the double hurdle model assumes that zero observations on the intensity equation might exist [61]. In contrast, the Heckman model assumes that there is no zero observation on the intensity model, once the decision to adopt has been made. In this study, Heckman's two-stage model was employed with the assumption that once the farmer had adopted quality seeds, there was no possibility of not allocating a piece of land for these seeds. The model has also been used in adoption studies because it effectively addresses sample selection bias [61,63].

The Heckman model jointly analyzes the two decisions: first, the adoption decision is modelled with a Probit model, and the inverse Mills ratio (IMR) is computed [64]. The model can be expressed as a latent variable model.

$$Y_i^* = \beta X_i + e_i \tag{1}$$

Where the observed variable $Y_i$ satisfies

$$Y_i = \begin{cases} 1 \text{ if } Y_i^* > 0 \\ 0 \text{ if } Y_i^* \leq 0 \end{cases} \tag{2}$$

The model can be empirically written as

$$Y_i = \beta X_i + e_i \tag{3}$$

Where: $Y_i$ is a binary variable taking the value of 1 if i$^{th}$ farmer adopts and 0 otherwise, $X_i$ are explanatory variables influencing adoption, $\beta$ is the coefficients to be estimated, $e_i$ is error term assumed to be normally distributed.

The second stage of the model estimates the intensity of quality seed adoption. This outcome is only observed for households that adopted quality seeds ($Y_i = 1$), and not for non-adopters ($Y_i = 0$). In other words, the zero values for non-adopters are not actual observations, but are missing by design [65]. As a result, the intensity equation was estimated using a linear regression model, with the Inverse Mills Ratio (IMR, λ) from the first-stage selection model included to correct for potential selection bias [63] as shown in equation 4.

$$Zi = \alpha X_i + \gamma \lambda + \varepsilon_i \tag{4}$$

Where: $Zi$ is the intensity of adoption of quality seeds, $\alpha$ are parameters to be estimated for the explanatory variables used, $\varepsilon_i$ is the error term, $\lambda$ is the inverse Mills ratio (lambda) and $\gamma$ is the coefficient to be estimated for $\lambda$. If the value is significant, it suggests selection bias, and the chosen Heckman model is the best fit [64].

## Pre-estimation checks for Heckman

We conducted several pre-estimation checks for the Heckman model to ensure robust results. First, we tested for sample selection bias by comparing the ordinary least squares (OLS) estimates with the Heckman model results of the outcome equation. A significant difference between the two suggests that selection bias existed (S1 Table). Secondly, we tested for the relevance of the two instruments used to determine whether they were correlated with selection and not the outcome equation. In this study, membership in the farmers' organization and access to input subsidies were used as instruments since they are reported to directly influence technology adoption by reducing transaction and production costs, but they do not substantially affect intensity once adoption occurs [66–68]. We also tested for over-identification by including the two instruments in the outcome equation to ensure that they did not significantly influence the intensity of adoption of quality seeds (S2 Table). The insignificant results supported the validity of their restriction exclusion. Thirdly, we checked for multicollinearity among independent variables using variance inflation factors (VIF), and a mean VIF of 3.92 was considered acceptable (S3 Table). Finally, we checked the distribution of residuals from the outcome equation using residual histograms with standard curves, kernel density plots, quantile-quantile (Q-Q) plots, and probability-probability (P-P) plots to test for significant variation from the joint normality assumption, as shown in (S1-S3 Figs). The shape of the four plots indicates that the joint normality assumption was not violated.

## Propensity score matching (PSM) model

To assess the impact of adopting quality seed potatoes on household food security, propensity score matching (PSM) was utilized because randomization was not feasible in our study. PSM is a non-parametric statistical method that estimates causal treatment effects in non-randomized studies by reducing selection bias from observable covariates [69,70]. It pairs adopters and non-adopters with similar observed characteristics, thus mimicking some randomized controlled trial (RCT) conditions within an observational study context [58,69]. Unlike other regression-based parametric models, the model is preferred because it does not require assumptions about the distribution of error terms or impose a specific functional form on the outcome equation, thereby reducing sensitivity to model misspecification [71]. The model is widely used in impact studies [60,72] to minimize selection bias from observable covariates and to estimate the average treatment effect (ATE) of technology adoption.

In this study, the model was used to analyze the impact of adopting quality seeds under the assumptions of conditional independence (unobservable factors do not influence quality seed adoption) and common support (significant overlap of propensity scores between adopters and non-adopters of quality seeds) [58,71]. PSM examines the control group and aims to identify individuals who are similar to the adopters (using selected observed covariates), then compares the outcome variable between the two groups, and if there is a difference, it is attributed to the technology [71]. This implies that, after controlling for all pre-intervention observable characteristics correlated with adopting quality seeds and the outcome variables, farmers who had adopted quality seeds had similar average outcomes as non-adopters would have if they had adopted. The differences in outcomes between the two groups were attributed to the adoption of quality seeds.

We implemented the PSM model by first regressing the dummy variable of adoption of quality seeds (D) on a set of household observable characteristics (X), using a probit model to generate households' propensity scores [73]. The model assumes a normal distribution of the error terms. The propensity score is specified as shown in equation 5.

$$Y = P(D_i = 1|X) \qquad (5)$$

Where, $Y$ is the propensity score, D is the treatment (D=1 for quality seed adopters), $X$ is a set of observable characteristics (Table 4).

We then used these scores to match the two groups (adopters and non-adopters) using nearest neighbor matching (NNM), kernel-based matching (KBM), and radius caliper matching (RCM). These matching estimators are widely used

because they produce similarly unbiased estimates [60]. Furthermore, using more matching estimators is advisable, given that PSM is sensitive to specification and matching method choice [73].

After matching, we imposed the common support region to ensure comparability, and only observations within this region were included. Covariate balance was assessed before and after matching using Pseudo R2, standardized mean bias, Rubin's B, and R statistics. The average treatment effect on the treated (ATT) was then calculated to evaluate the impact of quality seed adoption on household food security, measured using HDDS and FCS. ATT is regarded as a more suitable indicator for assessing the appropriateness of an intervention within a smallholder context than the population-wide Average Treatment Effects (ATE) [70,73]. The ATT was estimated as shown in equation 6.

$$ATT = E(Y1 – Y0|D = 1)$$ (6)

Where ATT is the average treatment effect on the treated, Y1 represents the HDDS and FCS of adopters and Y0 represents the HDDS and FCS for non-adopters.

## Results and discussion

### Descriptive statistics

**Characteristics of the sample based on the adoption of quality seeds.** The descriptive statistics of the sampled households based on their adoption behavior are presented in Table 1. The findings indicate that of the 541 sampled

**Table 1. Descriptive characteristics of the sample based on the adoption of quality seed.**

| Variable | Overall n = 541 | Adopters n = 239 | Non-adopters n = 302 | Difference in point |
|---|---|---|---|---|
| Age of household head (mean) | 43.34 | 41.53 | 44.78 | −3.25*** |
| Gender-male (%) | 73.57 | 74.90 | 72.52 | 2.38 |
| Pre-primary education (%) | 4.62 | 1.26 | 7.28 | −6.03*** |
| Primary education (%) | 35.49 | 23.01 | 45.36 | −22.35*** |
| Secondary education (%) | 46.77 | 56.90 | 38.74 | 18.16*** |
| Tertiary education (%) | 13.12 | 18.83 | 8.61 | 10.22*** |
| Household size (mean) | 4.05 | 4.00 | 4.08 | −0.08 |
| Extension access (%) | 30.87 | 42.26 | 21.85 | 20.41*** |
| Credit access (%) | 16.45 | 17.15 | 15.89 | 1.26 |
| Membership in farmers' organization (%) | 18.67 | 25.52 | 13.26 | 12.28*** |
| Training pesticide (%) | 35.67 | 53.97 | 21.19 | 32.78*** |
| Digital information (%) | 54.34 | 66.11 | 45.03 | 21.08*** |
| Access subsidies (%) | 37.71 | 42.28 | 30.13 | 17.15*** |
| Total land size (mean) | 1.09 | 1.34 | 0.90 | 0.44*** |
| Farming experience (mean) | 15.87 | 14.01 | 17.34 | −3.32*** |
| Livestock portfolio (mean) | 2.16 | 2.43 | 1.96 | 0.46*** |
| Registered farmers (%) | 15.34 | 18.41 | 12.91 | 5.50** |
| Distance seed sources (mean) | 4.04 | 3.45 | 4.50 | −1.05*** |
| Distance market (mean) | 6.11 | 5.10 | 6.91 | −1.81*** |
| Meru (%) | 29.57 | 40.59 | 20.86 | 19.72*** |
| Nyandarua (%) | 36.78 | 41.42 | 33.11 | 8.31** |
| Nakuru (%) | 33.64 | 17.99 | 46.03 | −28.03*** |

**Notes:** *p < 0.10, **p < 0.05 and ***p < 0.01: following t-test for difference in mean and a z-test for equality of proportions.

households, 239 farmers (44.18%) reported using quality seeds (adopters), while 302 farmers (55.82%) relied on uncertified seed sources (non-adopters). The adopters were male, younger, and better educated than non-adopters, who were female, older, and less educated. This implies that younger and more educated farmers are able to weigh the benefits and better understand new technology. They are also more willing to take risks, unlike older, risk-averse farmers. Male farmers are more likely to use quality seeds than female farmers. Female farmers face structural barriers that limit their access to resources and decision-making regarding adopting new technologies. The findings also demonstrate that adopters were more likely to access supportive institutions and resources than non-adopters. For instance, adopters were more likely to access extension services, agricultural training, group membership, and digital information. Moreover, these farmers were formally registered, a prerequisite for accessing government support such as fertilizer subsidies. This pattern highlights the role of institutional factors in enhancing the adoption of quality seeds by providing farmers with information and institutional support. In contrast, non-adopters were less likely to access this support, thus limiting their capacity to invest in quality seeds.

The results also indicate that non-adopters were remotely located, hindering their access to crucial agricultural inputs, markets, and government services compared to adopters, who travelled shorter distances to access these services. Geographic disparities in the adoption of quality seeds were evident, with most adopters coming from Meru and Nyandarua counties, while a larger proportion of non-adopters resided in Nakuru County. This can be attributed to the well-established seed production units and firms in Meru and Nyandarua counties, making it easier for farmers to access seeds. Conversely, farmers in Nakuru County had to travel to other counties to acquire seeds.

## Sources, recycling frequency, and varieties of quality seed potatoes

Table 2 illustrates regional differences in the adoption and sources of quality seeds, with Meru County exhibiting the highest adoption rate (60.62%), followed by Nyandarua County (49.75%). Nakuru County recorded the lowest adoption rate (23.63%). Additionally, there were variations in the access and distribution networks of seeds among the three counties. Most farmers, particularly those in Meru County (95%), sourced quality seeds from Kisima LTD, possibly due to the firm's dominance and proximity to farmers within the county, which was crucial in reducing the transaction costs of obtaining the seeds.

In contrast, farmers in Nyandarua County primarily sourced their quality seeds from multiple sources, such as research institutions (such as KARLO), private companies (such as FreshCrop), and local multipliers (such as Agricultural Training Center (ATC) and Tumaini National Youth Service [NYS]). The government established and funded these institutions to help farmers access quality seeds. Despite this, the quantity of quality seeds produced by these institutions was insufficient to meet the farmers' demand, as noted during KII, "*when you go to buy seeds from ATC or Tumaini, they claim they are already booked for the next two years. That is why you will find our farmers going to Kisima firm in Meru to buy quality seeds*" (SCAO Nyandarua County). A female farmer echoed the same concerns during the FGD meeting: "*Our farmers get seeds from Kisima Firm, but most of the time the firm is unable to meet our demand as it gets orders from farmers from other counties like Nyandarua who buy in bulk, making our local farmers not to access the seeds.*"

The same pattern was observed in Nakuru County, where farmers relied on several options, mainly FreshCrop, Molo Agricultural Development Corporation (ADC), and local multipliers, to obtain their seeds. However, the seed demand exceeded supply, causing farmers to travel to other counties to buy quality seeds. Farmers (12% in Nyandarua and 5% from Nakuru) could travel all the way to Kisima Ltd in Meru County to buy seeds. The shortage of quality seeds reported from these sources across the three counties indicates existing structural and systemic barriers to adopting quality seeds, as shown in Table 3. Besides, a higher rate of seed recycling further compounded this situation across the counties. On average, farmers in Nyandarua and Nakuru recycled seeds at least three times, while those in Meru County recycled them twice before changing the seed lot. The higher recycling rate among the three counties highlights accessibility issues to quality seeds and may also be a strategy to lower transaction costs associated with procuring these seeds.

**Table 2. Sources, frequency of recycling, and varieties of quality seed potatoes used.**

|  | Meru % | Nyandarua % | Nakuru % | χ2 |
|---|---|---|---|---|
| Adoption (Yes) | 60.62 | 49.75 | 23.63 | 51.23*** |
| **Sources of quality seeds** | % | % | % |  |
| Kisima | 94.85 | 12.12 | 4.65 | 169.33*** |
| Molo ADC firm | 0 | 7.07 | 25.58 | 28.05*** |
| Nakuru seeds | 1.03 | 0 | 4.65 | 5.30* |
| Freshcrop | 0 | 30.30 | 39.53 | 41.58*** |
| Agrovet | 5.15 | 1.01 | 2.33 | 3.03 |
| KARLO | 0 | 20.20 | 6.98 | 23.41*** |
| AGRICOS | 2.06 | 3.03 | 0 | 1.34 |
| Local multiplier | 7.22 | 45.45 | 23.26 | 37.49*** |
| **Frequency of recycling the seeds** |  |  |  | **F-test** |
| Frequency (mean) | 1.99(1.13) | 3.38(0.96) | 2.93(1.01) | 39.26*** |
| **Main varieties grown** | % | % | % | χ2 |
| Asante | 13.13 | 0 | 0 | 52.03*** |
| Markis | 1.88 | 7.04 | 0 | 16.66*** |
| Shangi | 76.29 | 83.84 | 100.0 | 12.27*** |
| Sherehekea | 28.13 | 0.50 | 0 | 112.47*** |
| Unika | 16.25 | 5.53 | 1.10 | 30.55*** |
| Others | 4.38 | 10.05 | 3.30 | 8.75** |
| N | 97 | 99 | 43 |  |

**Note**: *p<0.10, **p<0.05 and ***p<0.01.

**Table 3. Challenges hindering the adoption of quality seed potatoes.**

| Challenges | Meru% | Nyandarua % | Nakuru% | χ2 |
|---|---|---|---|---|
| Unavailability of seeds | 60.82 | 51.52 | 72.09 | 5.47* |
| Inaccessibility of seeds | 47.42 | 30.30 | 11.63 | 18.01*** |
| High costs | 21.65 | 54.55 | 13.95 | 32.97*** |
| Lack of preferred variety | 43.30 | 3.03 | 6.98 | 55.11*** |
| Low quality seeds | 9.28 | 9.09 | 0 | 4.27 |
| Demand not met | 18.56 | 9.09 | 13.95 | 3.69 |
| Distance to seed sources (mean) | 2.85(2.83) | 3.28(2.67) | 5.19(5.57) | 7.17*** |
| N | 97 | 99 | 43 |  |

**Notes:** *p<0.10, **p<0.05 and ***p<0.01.

The results further indicate that farmers in the three counties were variety-specific, with most preferring Shangi primarily due to its early-maturing attributes. This was noted during a key informant interview: "*Farmers here prefer Shangi because it is early maturing, has short dormancy, is easier to manage, and requires you to only spray it once per month*" (KII Buuri Sub-County). The varieties grown were also influenced by existing market demand. For instance, Markis was cultivated by farmers contracted by a processing company due to its good quality for making chips and crisps. Farmers in Meru County also produced other varieties such as Unika and Sherehekea. Unika was preferred due to its long dormancy

and storage quality. Sherehekea has delayed maturity and was not liked by many farmers; it was one of the varieties produced in Kisima Firm, so farmers had to grow it when other varieties were unavailable in the Firm.

## Barriers to access to formal seed systems

The main challenges preventing farmers from adopting quality seeds varied across regions, as shown in Table 3. However, the unavailability and inaccessibility of quality seeds in the three counties emerged as the prime barriers hindering farmers from making this adoption. Indeed, farmers, particularly those in Nakuru County, had to travel longer distances (5.19 km) to purchase quality seeds. Additionally, farmers expressed dissatisfaction with the seed rationing imposed by seed suppliers, which limited them to only two bags, far below what they required. To address the shortfall, they resorted to using uncertified seeds, as noted by views from KIIs/FGD: "*In this area, the majority of farmers use their own-saved seeds due to the unavailability of quality seeds*" (KII, Nyandarua County).

Another significant barrier to adoption was the high cost of seeds. Farmers perceived these seeds as costlier than the ones they bought from informal sources, thus demotivating them from using them. Further, most farmers voiced dissatisfaction with the size and variety of the seeds they received. Although most farmers preferred smaller-sized seed varieties, in many cases, such seeds were not available, as one participant remarked, "*When we order seeds, they sell us what is available and not what we actually need. For instance, when we order grade 1 (small-sized seeds: planted in a larger area and higher yield), we only get grade 2 (medium-sized seeds: grown in a smaller area and lower yield)*" (Lead farmer, Abothuguchi West Ward). This highlights a disconnect between seed suppliers and end-users, highlighting the necessity for a participatory approach in enhancing distribution networks.

## Household dietary diversity and quality

The household food and nutritional security was assessed using HDDS and FCS indicators, as shown in Table 4. Before accounting for control factors, adopters of quality seeds reported higher dietary diversity and food consumption scores than non-adopters across all three counties. In Meru County, adopters had a higher HDDS (8.49) than non-adopters (7.35), with a statistically significant difference of 1.15. Similarly, Nakuru County's HDDS showed a notable difference, with adopters scoring higher (8.23) than non-adopters (7.51), with a significant difference of 0.75. In contrast, Nyandarua County showed minimal differences between adopters and non-adopters, with no statistically significant difference observed for either HDDS or FCS. Examining nutrient-dense food categories, the intake of animal-based protein and energy-dense foods was significantly higher among Meru and Nakuru adopters than non-adopters. Conversely, no significant differences were seen between these food groups in Nyandarua. For micronutrient-rich foods, Meru adopters also reported higher intake (1.53) compared to non-adopters (1.16). No meaningful differences across counties were noted in plant protein-dense food consumption. FCS further reinforced this trend, with significant differences observed in Meru (7.42) and Nakuru (11.64), but not in Nyandarua. These findings suggest that adopting quality seeds is positively linked to improved dietary outcomes, although the impact may vary by location.

## Econometric results

**Determinants of the adoption and intensity of adoption of quality seeds.** The determinants of adoption and its intensity of quality seed potatoes were analyzed using the Heckman two-stage model, and the results are presented in Table 5. The significant coefficient of IMR ($\lambda = 0.423$, $p = 0.038$) confirms the selection bias in the outcome equation. More so, the significantly high correlation coefficient values ($\rho = 0.653$) imply that unobservable characteristics affected both the adoption and intensity of adoption of quality seeds. These findings suggest that the Heckman two-stage model was suitable and statistically robust in correcting selection bias and producing unbiased and valid estimates.

 

**Table 4. Household dietary diversity and quality.**

| Indicators | County | Adopters | Non-adopters | Difference |
|---|---|---|---|---|
| **HDDS** | Meru | 8.49 | 7.35 | 1.15** |
|  | Nyandarua | 7.73 | 7.59 | 0.14 |
|  | Nakuru | 8.23 | 7.51 | 0.72*** |
| **Nutrient-dense foods** |  |  |  |  |
| Animal protein | Meru | 1.67 | 1.16 | 0.51*** |
|  | Nyandarua | 1.39 | 1.20 | 0.19 |
|  | Nakuru | 1.77 | 1.40 | 0.37** |
| Plant protein | Meru | 0.75 | 0.76 | −0.01 |
|  | Nyandarua | 0.69 | 0.68 | 0.01 |
|  | Nakuru | 0.44 | 0.45 | 0.01 |
| Micronutrient | Meru | 1.53 | 1.16 | 0.37*** |
|  | Nyandarua | 1.46 | 1.48 | 0.02 |
|  | Nakuru | 1.26 | 1.22 | 0.04 |
| Energy | Meru | 3.58 | 3.32 | 0.26** |
|  | Nyandarua | 3.28 | 3.36 | −0.08 |
|  | Nakuru | 3.77 | 3.45 | 0.31*** |
| **FCS** | Meru | 81.80 | 72.41 | 9.39*** |
|  | Nyandarua | 72.15 | 68.85 | 3.30 |
|  | Nakuru | 84.49 | 74.19 | 10.29*** |

**Notes:** *p<0.10, **p<0.05 and ***p<0.01

The results indicate that age negatively influenced the probability of adoption and intensity of adoption of quality seeds. As farmers grow older, they are less likely to adopt new technology but tend to continue with their traditional ways of doing things. Education at all levels significantly influenced the adoption of quality seeds, but it did not substantially affect the intensity of adoption once the seeds were adopted. Similarly, access to extension significantly affected adoption but did not significantly affect the intensity of adopting quality seeds. Agricultural extension is crucial for promoting and disseminating information on new technologies, such as quality seed potatoes, thereby increasing the likelihood of adoption. Once farmers adopt, the intensity of quality seed adoption depends not on extension but on other factors. For instance, the results indicate that farmers with larger land sizes were more likely to increase the intensity of adoption of quality seeds. This can be attributable to the fact that farmers with larger farm sizes can devote some of it to experimenting with new seeds.

Interestingly, access to credit negatively influenced the adoption of quality seeds but increased the intensity of their adoption. This may imply that farmers who accessed credit might have diverted it to other uses, such as paying school fees. Nevertheless, once farmers adopted the seeds, access to credit enabled them to expand their usage. Results further indicate that access to digital information positively and significantly influenced the intensity of quality seed adoption. Digital tools such as mobile phones provide farmers with personalized information on crop management, weather updates, and market trends, thus motivating farmers to scale up the intensity of adoption of quality seeds.

Access to resources, such as livestock, significantly influenced both the adoption and the intensity of adoption of quality seeds. The number of livestock symbolizes wealth, which could enable farmers to adopt and expand the usage of quality seeds. In contrast, the distance to the market negatively influenced the adoption of quality seeds. Farmers farther from markets may incur higher transaction costs in accessing seeds and other agricultural inputs, which could offset their

**Table 5. Determinants of adoption and intensity of adoption of quality seed potatoes.**

| Variable | Selection model (probit) (Stage I: Adoption) n = 541 | | Outcome model (OLS) (Stage II: Intensity of adoption) n = 239 | |
|---|---|---|---|---|
| | Coef. | Z | Coef. | Z |
| Gender of household head | 0.102 (0.127) | 0.80 | 0.109 (0.082) | 1.33 |
| Age of household head | −0.016*** (0.005) | −3.02 | −0.010*** (0.004) | −2.69 |
| Primary education | 0.711*(0.401) | 1.77 | 0.289 (0.368) | 0.79 |
| Secondary education | 1.12***(0.404) | 2.76 | 0.336 (0.373) | 0.90 |
| Tertiary education | 1.097**(0.436) | 2.52 | 0.191(0.387) | 0.50 |
| Household size | −0.015 (0.038) | −0.38 | −0.014 (0.028) | −0.51 |
| Total land size | 0.081*(0.055) | 1.48 | 0.750***(0.032) | 23.63 |
| Total household income (KES) | 0.131*(0.070) | 1.88 | 0.119**(0.051) | 2.36 |
| Extension access | 0.327**(0.143) | 2.28 | 0.024 (0.105) | 0.23 |
| Credit access | −0.327*(0.179) | −1.82 | 0.270**(0.118) | 2.39 |
| Access high-value market | −2.49 (0.383) | −0.65 | 0.349 (0.249) | 1.40 |
| Digital information | −0.054 (0.138) | −0.39 | 0.155*(0.092) | 1.68 |
| Potato contract | 0.398 (0.539) | 0.74 | −0.455*(0.251) | −1.82 |
| Registered potato farmer | −0.23(0.191) | −0.12 | 0.118 (0.115) | 1.02 |
| Distance main road (km) | −0.066** (0.029) | −2.24 | 0.027 (0.115) | 1.02 |
| Distance seed sources | 0.030 (0.023) | 1.30 | 0.017(0.015) | 1.11 |
| Distance to market (km) | −0.023**(0.017) | 1.34 | −0.034*** (0.012) | −2.84 |
| Manure access | 0.199 (0.148) | 1.35 | 0.119 (0.097) | 1.22 |
| Livestock portfolio | 0.193***(0.056) | 3.42 | 0.095** (0.045) | 2.13 |
| Membership in farmers' organization | 0.397**(0.183) | 2.18 | | |
| Access input subsidies | 0.441***(0.134) | 3.30 | | |
| Nyandarua | −0.218*(0.168) | −1.30 | | |
| Nakuru | −0.601***(0.188) | −3.20 | | |
| Constant | −2.020**(0.797) | −2.54 | −1.795**(0.735) | −2.44 |
| Lambda | 0.423*(0.204) | 2.07 | | |
| Rho | 0.653 | | | |
| Sigma | 0.648 | | | |
| Wald chi2 (19) | 858.23 | | | |

**Notes**: Standard errors in parentheses. 1 US dollar (USD) = KES 133.03 in July 2022. * $p < 0.1$, ** $p < 0.05$, *** $p < 0.01$.

returns and discourage them from using quality seeds. Results further indicate that membership in farmers' organizations positively and significantly influenced the adoption of quality seeds. Groups create a platform where farmers can share information about new technologies, including quality seeds, allowing farmers to buy seeds in bulk, facilitate their transportation, access training within the group, and other resources, such as storage facilities, and benefit from collective marketing, which encourages adoption. Contrary to our expectations, farmers in the potato contract were less likely to intensify the adoption of quality seeds. However, the result confirms the views expressed during the FGD, where one farmer noted, *"Although farmers have been contracted, the majority do not sell through the contracts. The contracts buy at lower prices, e.g., when the potato was selling at 20-25 KES/kg, they were buying at 15-17 KES/kg. However, those who sell through them do so due to a lack of alternative markets"*.

Compared to Meru County, farmers in Nyandarua and Nakuru were less likely to adopt quality seeds, likely due to several challenges in obtaining them, such as unavailability and inaccessibility. This affirms the concerns expressed during KII/FGD that farmers from these two counties had to travel to other counties to purchase seeds, thereby increasing the transaction costs of obtaining them, thereby discouraging their adoption.

**Post-estimation checks for Heckman results.** In addition to the likelihood ratio test and inverse Mills ratio described in section 4.2.1, we conducted robustness checks by comparing Heckman estimates with alternative models (probit with truncated regression and probit with OLS). The results, particularly on the main variables, were consistent with those of the Heckman Model, further strengthening the validity of the results (S4 Table).

**Impact of quality seed adoption on household food security.** Table 6 shows how adopting quality seed potatoes affects two key dietary quality indicators (HDDS and FCS). The probit model was first used to calculate propensity scores. After matching, the balancing property was satisfied, as shown in Table 6. The LR test was insignificant (p = 0.883), and the Pseudo R2 was very low (0.024), indicating successful matching between the treated and control groups. The standardized mean and median biases were below 25% thresholds, also affirming good balance. Additionally, Rubins' R (0.84) was within acceptable ranges (0.5–2), confirming that the matching process achieved good covariate balance. Further, the common support condition was satisfied, and there was sufficient overlap between the propensity score distribution for adopters and non-adopters (Fig 2).

After confirming that propensity score matching was successful (Table 6), the impact of quality seed adoption on household food security was estimated using KBM, NNM, and RCM with a caliper of 0.05. The findings show that adopting quality seeds had a positive, statistically significant effect on HDDS, with this effect remaining consistent across all methods. Specifically, adopters of quality seeds had an average HDDS of 8.10 against 7.62 for non-adopters. The ATT for using quality seeds ranged from 0.49 to 0.53, indicating that adopting quality seeds contributes significantly to household food security and nutrition. Similarly, the average treatment effect of adopting quality seeds on FCS remained positive and statistically significant across all three matching approaches. The estimated differences ranged from 6.02 to 6.24 points, with all results significant at the 1% level. These findings suggest that quality seed adoption is associated with improved household dietary diversity and overall food consumption quality.

**Table 6. Effect of quality seed adoption on dietary diversity and food consumption scores.**

| Variable | Sample | Treated | Control | Difference | S.E. | T-stat | |
|---|---|---|---|---|---|---|---|
| **HDDS** | Unmatched | 8.13 | 7.50 | 0.63*** | 0.14 | 4.36 | |
| NNM | ATT | 8.10 | 7.57 | 0.53** | 0.22 | 2.41 | |
| KBM | ATT | 8.10 | 7.61 | 0.49** | 0.20 | 2.40 | |
| RCM | ATT | 8.10 | 7.62 | 0.49** | 0.20 | 2.38 | |
| **FCS** | unmatched | 78.29 | 72.05 | 6.23*** | 1.49 | 4.17 | |
| NNM | ATT | 78.38 | 72.14 | 6.24*** | 2.17 | 2.87 | |
| KBM | ATT | 78.38 | 72.31 | 6.07*** | 2.09 | 2.90 | |
| RCM | ATT | 78.38 | 72.36 | 6.02*** | 2.09 | 2.88 | |
| **Matching quality** | | | | | | | |
| Sample | Ps R2 | LR chi2 | meanBias | MedBias | B | R | %var |
| Unmatched | 0.225 | 166.79*** | 30.1 | 32.0 | 119.6 | 0.68 | 75 |
| After-matching | 0.024 | 15.31 | 5.4 | 4.4 | 36.7 | 0.84 | 75 |

**Note:** *p < 0.10, **p < 0.05 and ***p < 0.01.

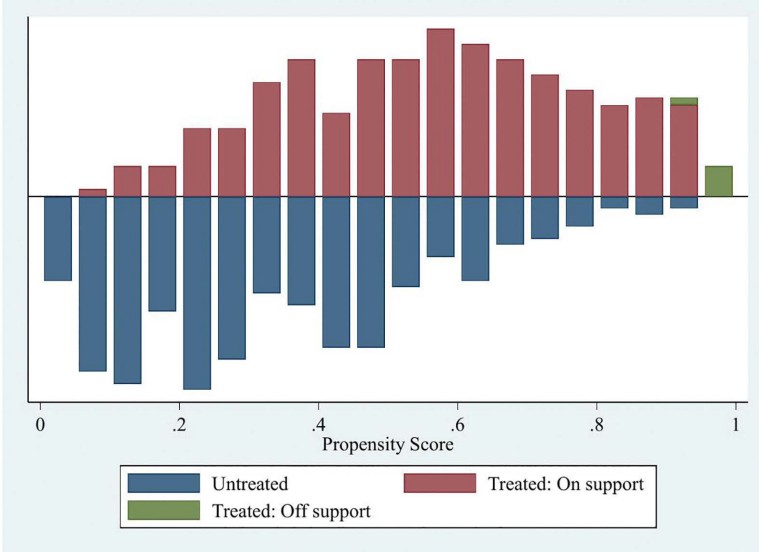

**Fig 2. Propensity score distribution and common support.**

## Sensitivity analysis

Sensitivity analysis was conducted to check the robustness of the result using the Rosenbaum bounds test (rbounds). This helped to determine whether any hidden bias could alter the inferences of the outcome [74]. The results (S5 Table) indicate that the estimated effect of the treatment on HDDS and FCS is robust to hidden bias. At gamma ($\Gamma = 1$), the effect was significant (sig+=sig-=0), and the significance remained constant even after allowing gamma to increase from 1 to 2 ($\Gamma = 2$, sig+=sig-=0). The Hodges-Lehmann point (t-hat+) indicates only a slight decline as the gamma value doubled (from 8 to 7.5 for HDDS and 75.5 to 69.75 for FCS), implying that the positive effect of the treatment on both indicators of food security was solely due to the adoption of quality seeds and not unobserved factors.

## Discussion

Despite concerted efforts by the government and development partners to promote quality seed potato technologies in Kenya, the adoption rate remains relatively low. This study found that only 44.18% of smallholder potato farmers adopted these seeds in the study area, with inaccessibility and high costs cited as the main challenges. Besides the supply chain inefficiencies, such as inadequate seed multiplication capacity, limited distribution networks, and the unavailability of pre-ferred varieties, further impede the adoption. In particular, seed companies could not meet demand during peak planting periods, thus prompting farmers to resort to informal sources or recycle seed stocks more than three times. While recycling may be perceived as a way to reduce the production cost of using seeds, higher recycling can increase vulnerability to diseases and pests, and decrease crop yields [28]. These findings corroborate the previous findings [25,32,37], who also documented low adoption rates ranging between 2% to 50% and higher recycling rates of up to 5 times, highlighting institutional and systemic constraints in potato seed systems in Kenya.

Interestingly, while the majority of non-adopters across the three counties were aware of the risks and challenges of using uncertified seed, such as spreading pests and diseases (85.2%), decline in yields (79.4%), and reduced ware quality (36.6%), the vast majority continued to rely on informal seed sources (own-saved, farmer-to-farmer exchange, local markets or seed-for-seed exchange). These farmers believed that rotating seeds with those from different geographical

areas (such as exchanging seeds across different sub-counties) was a way of renewing seeds. For instance, during FGD, some farmers stated that shifting seed sources between regions was sufficient to restore productivity. One farmer at Imenti Central Sub-County noted that "*you cannot plant the seeds from this area, otherwise you will not harvest anything; farmers here always purchase seeds from Buuri Sub-County because seeds from there do well here*". This statement shows that most farmers associate location or field change with seed quality renewal, rather than recognizing the underlying problems of seed degeneration due to viruses in the tubers themselves. These findings align with misconceptions around seed renewal and degeneration reported by [37]. They partly explain why many farmers recycle seed, thereby contributing to lower adoption rates of quality seeds [43].

Econometric results highlight the significant role of socio-economic and institutional factors in influencing both the adoption and intensity of adoption of quality seed potato. Analyzing socio-economic factors, our findings show that education at all levels positively and significantly affected adoption, but it did not significantly impact the intensity of adoption of quality seed. This suggests that education plays a key role in the initial decision to adopt, as it enhances farmers' awareness and understanding of the benefits of quality seeds. However, once adoption occurs, education level does not affect how much they use these seeds, suggesting that other factors more strongly determine adoption intensity. These findings are consistent with [51], who also found that education influences the decision to adopt agricultural technology but does not affect its level of adoption.

In line with [75], we observed that farmers with larger landholdings are more likely to allocate more land to quality seeds, unlike those with smaller plots. Often, land is used as collateral to access loans, helping farmers overcome liquidity constraints when acquiring and using quality seeds. Moreover, larger landowners benefit from economies of scale, such as bulk purchasing of quality seeds. This improves their access to seed multipliers who, due to demand, tend to sell seeds in large quantities. Previous studies have linked larger land size to reduced production costs, as it allows transaction costs to be spread across more acres, making new technologies like quality seeds more economically attractive [75,76]. Total household income also played a crucial role in both adoption and intensity decisions. Higher income enables farmers to purchase quality seeds and other complementary inputs such as fertilizers and agro-chemicals. This finding aligns with [50], who stated that higher-income farmers can invest in capital-intensive technologies like quality seeds, which require more fertilizers, agrochemicals, and management. Siyum et al. [51] also emphasized the role of household income in alleviating liquidity constraints faced by smallholders, especially where credit markets are limited.

Further results indicate that farmers with large numbers of livestock and those accessing input subsidies were more likely to adopt and increase the intensity of quality seeds adoption. Livestock often serve as a proxy for wealth and asset endowment [77]. More livestock allows farmers to ease financial constraints by selling livestock products and byproducts such as milk. Higher livestock numbers can also generate more organic manure, substituting commercial fertilizer, lowering production costs, and increasing potential returns. On the other hand, subsidies reduce production costs, motivating farmers to adopt quality seeds. These findings support previous studies that found a positive relationship between asset ownership and the adoption of quality seed technologies [49,50].

Regarding institutional factors, the findings show that farmers with access to extension agents were more likely to adopt quality seeds than those without such access. However, once adoption occurred, access to extension agents did not influence the amount of land allocated to these quality seeds, suggesting that extension agents are key in disseminating and promoting new technology at the initial stage. Once adoption occurs, the intensity of adoption is determined by other factors. The results agree with [78], who stated that extension agents are crucial during the early stages of technology adoption because they help transmit information about the technology's attributes to farmers, but do not determine how intensively these technologies are used.

Membership in farmer organizations plays a key role in strengthening seed systems and overcoming structural barriers to quality seeds. Some groups were farmer-initiated, while others, such as village models, were formed with external support from extension agents or NGOs such as KOPIA, a Korean-funded organization. The main objective of these groups is

to help farmers buy and transport seeds in bulk, which is essential given that commercial seed multipliers such as Kisima Ltd. often sell seeds only in large quantities (200 bags or more), beyond the reach of most resource-constrained small-holders. Through collective action, farmers significantly reduce barriers to accessing quality seeds by lowering acquisition costs. For instance, farmers pay about KES150 per bag for transportation, compared to KES600 when buying individually. Beyond access and cost reduction, farmer organizations also provide training in seed multiplication, initial capital for seed purchases, and diffused-light storage facilities, enabling members to multiply and sell clean seeds to other farmers. This aligns with findings by [79], who emphasized the role of groups in providing farmers with information on new technologies, markets, and resources such as credit, all of which promote their adoption. Moreover, farmers can negotiate better prices and leverage economies of scale to sell their produce through their groups [80,81].

Interestingly, access to credit had a negative effect on the initial adoption of quality seeds but significantly increased the intensity of use once these seeds were adopted. This credit paradox suggests that liquidity alone may not be the primary constraint at the initial stage of technology adoption. Other factors, such as risk perception, agronomic uncertainty, and competing household priorities (such as health, school fees, and smoothing consumption), may play a more central role in early adoption decisions, whereas the intensity of adoption appears to be more capital-dependent [82]. These findings are consistent with experimental evidence from Tanzania showing that access to microcredit does not necessarily increase agricultural technology adoption or productivity [83]. Similarly, [84] attributes the negative effect of credit access on the adoption of improved sorghum varieties in Kenya to risk perception, arguing that when farmers perceive high production risks, they may be reluctant to invest borrowed funds in improved technologies. These findings highlight the importance of bundling credit with other agricultural inputs or providing it in kind for effective use.

The study also highlighted the evolving role of digital information. Contrary to previous studies that have shown a positive relationship between digital information and the adoption of new agricultural technologies [85,86], our findings indicate that access to digital platforms had no significant effect on initial adoption, possibly because adoption decisions are still primarily driven by personal networks, extension agents, and physical seed access. However, once farmers have adopted quality seeds, access to digital information becomes critical in determining how intensively they use them. Digital platforms offer timely, personalized information on farming practices, weather forecasts, pest and disease alerts, and market prices [68]. This information helps farmers make better decisions, leading to higher yields and increased production. These findings support previous studies indicating that digital innovations effectively complement conventional extension services rather than replacing them [87,88].

Other significant barriers included age, distance to the main road, and distance to markets. As farmers grow older, they tend to stick to traditional practices and are less likely to try innovations. The findings align with [49,89], who suggested that farmers become skeptical of new technologies as they age. Likewise, farmers farther from the main road and markets face higher transaction costs when accessing markets or improved seeds, discouraging the adoption of quality seeds and promoting reliance on locally sourced seeds from neighbors or their own stored seeds. These spatial and demographic barriers are a pointer to the need for localized input delivery systems and tailored interventions that are friendly to older farmers.

The impact of quality seeds on food and nutritional security, as reflected by HDDS and FCS, was evident. The adopters of quality seeds reported an average increase of 0.49 in their HDDS and 6.09 in their FCS, indicating greater access to diverse, nutritious foods than non-adopters. This demonstrates that adopting quality seeds can directly improve food and nutritional security through increased yields, which boosts food availability at the household level and allows farmers to access a variety of food groups. It can also indirectly enhance food security through increased market surpluses, generating income, especially when markets are available, thereby boosting household purchasing power and access to a wide range of food items. The results align with existing literature [2,14], which links agricultural technology to better nutritional outcomes. The study, therefore, emphasizes the potential of strengthening seed systems as a pathway to achieving food and nutritional security.

## Conclusions and policy implications

The study analyzed the challenges and drivers of the adoption and intensity of quality seed potato use among smallholder farmers. The key findings indicate that the main challenges hindering potato farmers from using quality seeds are their unavailability, inaccessibility, and high cost. Moreover, a lack of preferred varieties limits farmers' adoption of these seeds. The main factors influencing both the adoption and intensity of adoption of quality seed potatoes were age, land size, total income, and distance to market. Other factors, such as digital information and credit access, were found to influence the intensity of adoption but did not significantly influence initial adoption, suggesting that digital advisory services enhance input intensification rather than initial adoption. Similarly, access to credit alone may not influence initial adoption, but it is crucial for scaling up the usage of quality seeds once they are adopted. Farmers who used quality seeds had higher HHDS and FCS, which translated into improved food and nutritional security. These findings underscore the importance of promoting the adoption of quality seed as a pathway to improving dietary diversity and quality among smallholder farmers.

These results have important policy implications. They emphasize the need for policymakers and development partners to support the creation of seed multiplication units in rural areas to ensure a steady supply of quality seeds. Additionally, many farmers, especially those organized into groups, should receive training and certification to multiply seeds, increasing availability and accessibility for smallholder farmers. Moreover, policymakers should strengthen traditional extension methods by integrating them with digital technologies to ensure that farmers, particularly those in resource-constrained areas and remote regions, can access timely, personalized information.

Additionally, policies should not only aim to promote initial adoption but also emphasize strategies that enhance continuous access, availability, and affordability of quality seeds among smallholder farmers. Interventions such as improved market access, sustained extension support, effective seed distribution networks, better access to seed subsidies, and secure land rights can strengthen long-term adoption of quality seeds. This, in turn, can contribute to improved food and nutritional security.

While this study offers valuable insights into the challenges and determinants of quality seed adoption, the intensity of adoption, and its impact on household food security, the cross-sectional nature of the data limits our ability to determine duration or long-term adoption dynamics, such as whether farmers consistently adopt quality seed each season or if adoption is temporal. This distinction is important because sustained use, rather than temporal adoption, is more likely to generate long-term productivity and welfare gains. Therefore, future research should employ panel or duration models to assess the sustainability of adoption. For example, they could examine the duration between awareness and adoption, and the persistence of quality seed use over time, to better understand sustainability dynamics

## Supporting information

**S1 Table. Comparison of Heckman with OLS estimates of the outcome equation (sensitivity analysis).**
(DOCX)

**S2 Table. Over-identification checks of the instruments used in the selection equation.**
(DOCX)

**S3 Table. VIF test for multicollinearity among independent variables in the outcome equation.**
(DOCX)

**S4 Table. Comparison of Heckman outcome coefficients with marginal effects from alternative models for robustness checks.**
(DOCX)

**S5 Table. Sensitivity analysis using Rosenbaum bounds for HDD and FCS.**
(DOCX)

**S1 Fig. Residual histogram with a normal curve and kernel density plots.**
(TIF)

**S2 Fig. Q–Q plot of Heckman residuals.**
(TIF)

**S3 Fig. P-P plot of Heckman residuals.**
(TIF)

## Acknowledgments

The authors are grateful to the research assistants and agricultural extension officers from the three counties for their support throughout the data collection period. Special thanks to the smallholder potato farmers for their willingness to participate and share their valuable insights during the survey.

## Author contributions

**Conceptualization:** Margaret N. Mwangi, Wilckyster N. Nyarindo, Marther W. Ngigi, Hezron N. Isaboke.

**Data curation:** Margaret N. Mwangi, Wilckyster N. Nyarindo, Hezron N. Isaboke.

**Formal analysis:** Margaret N. Mwangi, Wilckyster N. Nyarindo, Marther W. Ngigi, Hezron N. Isaboke.

**Funding acquisition:** Marther W. Ngigi.

**Investigation:** Margaret N. Mwangi, Marther W. Ngigi, Hezron N. Isaboke.

**Methodology:** Margaret N. Mwangi, Wilckyster N. Nyarindo, Marther W. Ngigi, Hezron N. Isaboke.

**Project administration:** Margaret N. Mwangi, Marther W. Ngigi.

**Supervision:** Wilckyster N. Nyarindo, Marther W. Ngigi, Hezron N. Isaboke.

**Validation:** Margaret N. Mwangi, Wilckyster N. Nyarindo, Marther W. Ngigi.

**Visualization:** Margaret N. Mwangi, Wilckyster N. Nyarindo, Marther W. Ngigi, Hezron N. Isaboke.

**Writing – original draft:** Margaret N. Mwangi, Wilckyster N. Nyarindo, Marther W. Ngigi, Hezron N. Isaboke.

**Writing – review & editing:** Margaret N. Mwangi, Wilckyster N. Nyarindo, Marther W. Ngigi, Hezron N. Isaboke.

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
