## [Decision Letter · Decision Letter 0]

28 Jan 2026

PONE-D-25-50752Seeding food security: Overcoming barriers to quality potato seed adoption among smallholders in KenyaPLOS One

Dear Dr. Isaboke,

Thank you for submitting your manuscript to PLOS ONE. After careful consideration, we feel that it has merit but does not fully meet PLOS ONE’s publication criteria as it currently stands. Therefore, we invite you to submit a revised version of the manuscript that addresses the points raised during the review process.

**ACADEMIC EDITOR:** Please see comments below

We look forward to receiving your revised manuscript.

Kind regards,

Charles Odilichukwu R. Okpala, PhD

Academic Editor

PLOS One

Journal Requirements:

2. In the ethics statement in the Methods, you have specified that verbal consent was obtained. Please provide additional details regarding how this consent was documented and witnessed, and state whether this was approved by the IR

5. We note that Figure 2 in your submission contain map images which may be copyrighted. All PLOS content is published under the Creative Commons Attribution License (CC BY 4.0), which means that the manuscript, images, and Supporting Information files will be freely available online, and any third party is permitted to access, download, copy, distribute, and use these materials in any way, even commercially, with proper attribution. For these reasons, we cannot publish previously copyrighted maps or satellite images created using proprietary data, such as Google software (Google Maps, Street View, and Earth). For more information, see our copyright guidelines: http://journals.plos.org/plosone/s/licenses-and-copyright.

1. You may seek permission from the original copyright holder of Figure 2 to publish the content specifically under the CC BY 4.0 license.

**Additional Editor Comments:**

Please, address concerns raised by reviewers in great detail. Thank you

Reviewers' comments:

Reviewer's Responses to Questions

**Comments to the Author**

1. Is the manuscript technically sound, and do the data support the conclusions?

Reviewer #1: Yes

Reviewer #2: Yes

2. Has the statistical analysis been performed appropriately and rigorously? 

Reviewer #1: I Don't Know

Reviewer #2: Yes

3. Have the authors made all data underlying the findings in their manuscript fully available?

Reviewer #1: Yes

Reviewer #2: No

4. Is the manuscript presented in an intelligible fashion and written in standard English?

Reviewer #1: Yes

Reviewer #2: Yes

5. Review Comments to the Author

Reviewer #1: 1. Clarity on “quality seed” classification

Quality seed includes certified seed, rapid multiplication, and “known-source” seed not formally inspected. This category is broad. The paper should clarify:

how many farmers used certified vs. non-certified quality seeds, and

whether impacts differ by seed type.

2. Need stronger explanation of intensity variable

Intensity is defined as land allocated to quality seed, but later model output mentions “acres planted” rather than “proportion” — this needs consistency and clearer measurement.

3. Some factors have unexpected signs

Credit access negatively influences adoption but increases intensity. This is explained, but it would be stronger if supported with:

more qualitative evidence, or

discussion of how credit is used (school fees, health, etc.)

4. Sustainability and long-term adoption not addressed

The study itself acknowledges it does not examine when adoption occurred or whether farmers will continue using quality seed. This is important for policy, since seed adoption may be seasonal or temporary.

Reviewer #2: Summary of the Research This manuscript investigates the factors influencing the adoption and intensity of use of quality potato seeds among smallholder farmers in Kenya (Meru, Nakuru, and Nyandarua counties). Using a sample of 541 households, the authors employ a Heckman two-stage model to analyze adoption drivers and Propensity Score Matching (PSM) to assess the impact of adoption on household food security. The study finds that while adoption rates are relatively low (approx. 44%), the use of quality seeds significantly improves household dietary diversity and food consumption scores.

General Assessment The manuscript addresses a critical issue regarding food security and agricultural productivity in Sub-Saharan Africa. The research question is clearly defined, and the study design is technically sound. The use of a multi-stage stratified sampling technique ensures a representative sample, and the choice of econometric models (Heckman and PSM) is appropriate for handling selection bias and analyzing the distinct phases of technology adoption.

However, there is a critical compliance issue regarding the Data Availability Policy that must be resolved before publication. Additionally, there are minor formatting and grammatical errors that require correction.

Specific Comments & Suggestions

1. Methodology and Analysis

Model Justification: The justification for using the Heckman two-stage model over Tobit or Double Hurdle models is well-reasoned and supported by the literature . The distinction that adoption and intensity are sequential decisions influenced by different factors is a strong point of the analysis.

Instrument Validity: The authors have rightly identified "membership in farmers' organizations" and "access to subsidies" as exclusion restrictions. The pre-estimation checks, including the over-identification test (S2 Table) and VIF analysis (mean 3.92), suggest the statistical approach is rigorous .

Robustness Checks: The use of three different matching algorithms (NNM, KBM, RCM) and the Rosenbaum bounds sensitivity analysis significantly strengthens the validity of the food security impact findings .

2. Results and Interpretation

Credit Access Paradox: The finding that credit access negatively influences initial adoption but positively influences intensity of use is fascinating . The discussion suggests this may be due to the diversion of funds to non-agricultural needs. It would be valuable if the authors could expand on this in the Discussion section—perhaps suggesting that initial adoption relies more on risk perception than liquidity, whereas scaling up (intensity) is purely capital-dependent.

Formatting of Units: There are several instances where LaTeX-style formatting codes have remained in the text (e.g., Page 13, line 131: $8~tons/ha$). Please strip these formatting tags to ensure the text is clean standard English.

3. Minor Corrections (Grammar and Typos)

Page 19, Lines 417-418: The phrase "This implies that young and more educated farmers can be able to weigh the benefits..." is tautological. Please revise to "are able to weigh" or "can weigh" .

Page 35, Line 639: "These findings align with misconceptions... reported by [37] and partly explain why..." strictly speaking, the subject "misconceptions" is plural, but if the subject is the "findings" (plural), "explain" is correct. However, if referring to the "study" or "report", check for subject-verb agreement.

Page 53 (References): There is a typo in the supporting information list: "Rosesnbaum bounds" should be corrected to "Rosenbaum bounds".

Consistency: Ensure consistent capitalization of "Sub-County" throughout the manuscript.

6. PLOS authors have the option to publish the peer review history of their article (what does this mean?). If published, this will include your full peer review and any attached files.

Reviewer #1: No

Reviewer #2: **Yes:** Guy Roussel Takuissu Nguemto

---

## [Author Response · Author response to Decision Letter 1]

21 Feb 2026

Dear Editorial Office,

We have addressed all reviewer and editor comments in detail in the attached Response to Reviewers document. All suggested revisions have been incorporated into the revised manuscript and supporting files. We believe the manuscript has been substantially improved as a result of this feedback.

Sincerely,

Hezron Nyarindo Isaboke (Corresponding author)

---

## [Decision Letter · Decision Letter 1]

24 Mar 2026

Seeding food security: Overcoming barriers to quality potato seed adoption among smallholders in Kenya

PONE-D-25-50752R1

Dear Dr. Isaboke,

We’re pleased to inform you that your manuscript has been judged scientifically suitable for publication and will be formally accepted for publication once it meets all outstanding technical requirements.

Kind regards,

Charles Odilichukwu R. Okpala, PhD

Academic Editor

PLOS One

Additional Editor Comments (optional):

Accepted for publication.

Reviewers' comments:

Reviewer's Responses to Questions

**Comments to the Author**

1. If the authors have adequately addressed your comments raised in a previous round of review and you feel that this manuscript is now acceptable for publication, you may indicate that here to bypass the “Comments to the Author” section, enter your conflict of interest statement in the “Confidential to Editor” section, and submit your "Accept" recommendation.

Reviewer #1: All comments have been addressed

2. Is the manuscript technically sound, and do the data support the conclusions?

Reviewer #1: Yes

3. Has the statistical analysis been performed appropriately and rigorously? 

Reviewer #1: Yes

4. Have the authors made all data underlying the findings in their manuscript fully available?

Reviewer #1: Yes

5. Is the manuscript presented in an intelligible fashion and written in standard English?

Reviewer #1: Yes

6. Review Comments to the Author

Reviewer #1: all comments and query have been addressed properly. suitable for publication. Grammatical and English writing also rechecked and corrected with highest priority.

7. PLOS authors have the option to publish the peer review history of their article (what does this mean?). If published, this will include your full peer review and any attached files.

Reviewer #1: No

---

## [Editor Report · Acceptance letter]

PONE-D-25-50752R1

PLOS One

Dear Dr. Isaboke,

I'm pleased to inform you that your manuscript has been deemed suitable for publication in PLOS One. Congratulations! Your manuscript is now being handed over to our production team.

Kind regards,

on behalf of

Dr. Charles Odilichukwu R. Okpala

Academic Editor

PLOS One